# Impact of Investment in Tourism Infrastructure Development on Attracting International Visitors: A Nonlinear Panel ARDL Approach Using Vietnam's Data

Quang Hai Nguyen [1,2] 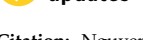

1 Faculty of Business Administration, University of Econmics and Law, Ho Chi Minh City 71309, Vietnam; nhquang@uel.edu.vn
2 Vietnam National University, Ho Chi Minh City 71309, Vietnam

**Abstract:** Investment in tourism infrastructure development to make destinations and services increasingly attractive is considered a key measure in developing a country's tourist destinations. This paper investigates the impact of investment in tourism infrastructure components on international visitor attraction using data from Vietnam for the period 1995–2019. The results of analyzing panel data by the nonlinear Autoregressive Distributed Lag (ARDL) approach show that, in the long-run, investing in the three components of tourism infrastructure, namely transport and communications infrastructure, the hotel and restaurant industry, and recreation facilities, has a strong and positive impact on international visitor attraction. In addition, different short-run impacts of the three tourism infrastructure components on the whole market and each major international visitor market are also found.

**Keywords:** tourism infrastructure; attracting international visitors; transport and communications infrastructure; hotel and restaurant industry; recreation facilities

## 1. Introduction

Tourism plays a vital role in the economic growth of many countries, contributing to the development of related services and infrastructure. Thus, the development of tourism affects the progress and prosperity of the national economy (Sinclair 1998). International tourists bring foreign currencies to destination countries, increase residents' incomes, create jobs, improve living standards, and contribute to expanding and strengthening international economic relations. Tourism development has become an important goal for most governments, especially in developing countries. Therefore, studying and proposing policies to develop tourism has become an issue of interest to both governments and researchers in recent years.

According to Boers and Cottrell (2007), the demands of tourists in the 21st century are very specialized and varied, so tourists are not simply satisfied with conventional travel experiences. To meet the unique and diverse demands of tourists, Dujmovic and Vitasovic (2014) argue that it is important to develop new tourism products and destinations, providing tourists with more sources of inspirational experience. Matias et al. (2007) point out that factors driving tourism's growth and development have been identified and improved, including improved income and wealth, improved traffic, changes in lifestyles and consumption values, entertainment space, international globalization, immigration, special events, education, information and communications technology, marketing, promotion of tourist destinations, infrastructure in general, and tourism infrastructure in particular. Therefore, it can be said that improving tourism infrastructure to increase the attractiveness of the destination is an essential factor in attracting tourists. The studies of Tribe (2004), Naudé and Saayman (2005), and Seetanah et al. (2011) point out that a country's infrastructure determines its potential attractiveness as a tourist destination.

Furthermore, recent studies have shown that tourism infrastructure has a positive impact both directly and indirectly on the quality of life of residents through sustainable tourism development (Mamirkulova et al. 2020). Therefore, there have been studies related to tourism infrastructure, although this issue is not always entirely the focus of research, such as those considering the role of infrastructure in tourism development (Prideaux 2000; Jovanović and Ilić 2016), infrastructure impact on tourism development (Seetanah et al. 2011; Yu 2016), the relationship between infrastructure and tourism (Suleiman and Albiman 2014; Mandić et al. 2018), the impact of transport infrastructure on tourism development (Khadaroo and Seetanah 2007a, 2007b, 2008; Seetanah and Khadaroo 2009; Ouariti and Jebrane 2020), relationship between tourism infrastructure and international visitor flows (Lim et al. 2019), and the relationship between foreign direct investment and tourism development (Selvanathan et al. 2012; Khoshnevis Yazidi et al. 2015; Samimi et al. 2017). These studies have shown the impact of infrastructure, or some of its components, on tourism development in various contexts. However, to the best of our knowledge, studies considering the full impact of tourism infrastructure components such as transport infrastructure, social infrastructure, and environmental infrastructure on attraction to tourists are rare. This is the driving force for this study, examining the role of investment in tourism infrastructure development and in attracting international tourists, using empirical data from Vietnam.

Vietnam is a developing country located in Southeast Asia with many historical relics and famous landmarks, notably including eight UNESCO heritage sites. The tourism industry plays a vital role in the development of the economy. Therefore, it is seen as a key economic sector. According to the Vietnam National Administration of Tourism, Vietnam National Administration of Tourism (2020), in 2019 the tourism industry directly contributed to 9.2% of Vietnam's GDP, including the vital role of international tourists. However, despite having diversified and abundant tourism resources, if investment in tourism infrastructure development is limited, Vietnam will become an unattractive tourist destination and will be unable to compete with regional destinations such as Thailand, Malaysia, or Singapore.

This study aims to determine the impact of investment in tourism infrastructure development on attracting international tourists. The important contribution of this paper will be a detailed description of the different roles of investment in transport and communication infrastructure development, the hotel and restaurant industry, and recreation facilities in attracting international tourists, with an updated sample to 2019. Research results are expected to contribute both theoretically and practically, providing necessary implications to attract future tourism development investment.

After the introduction section, the structure of the study includes four further sections: Section 2 presents a literature review; Section 3 presents the methodology and data; Section 4 presents the research results and discussion; and finally the article ends with the conclusion in Section 5.

## 2. Literature Review

Tourism is viewed as one of the fastest growing fields over recent decades, especially in emerging and developing economies. According to Thapa (2012), although the tourism industry has obviously grown, it is important to maintain and develop it with a sustainable strategy for further expansion. Investment in infrastructure development, emphasizing tourism infrastructure, is considered one of the critical factors to help achieve this goal. Scholars and policymakers agree that infrastructure development plays a key role in maintaining visitor arrivals and overall economic growth (Suleiman and Albiman 2014; Yu 2016). "The maintenance of local tourism infrastructure is becoming an increasingly important prerequisite for the country's competitiveness" (Petrova et al. 2018, p. 259). Moreover, widespread and efficient infrastructure is an important factor in ensuring the efficient functioning of the economy (Bookman and Bookman 2007). Conversely, weak infrastructure can

disrupt a country's economic development and international competitiveness (Tribe 2004; Hope 2010).

Tourism infrastructure is a type of infrastructure consisting of facilities and services performed within a particular locality to meet the needs of local residents and specific purposes (Goeldner and Ritchie 2009). "It is considered as the physical element that is designed and erected to cater to visitors" (Jovanović and Ilić 2016, p. 289). Tourism infrastructure has the potential to increase competition and promote tourism by providing travel facilities for tourists (Suleiman and Albiman 2014). Further, Lim et al. (2019, p. 187) pointed out that "tourism infrastructure increases tourism demand trends". The scope of tourism infrastructure is wide and involves all the factors that can facilitate and promote a destination's tourism development (Swarbrooke and Horner 2001). In a broad sense, tourism infrastructure encompasses all of the means that tourists use when they leave home, arrive at their destination, and return home (Lohmann and Netto 2017).

Tourism infrastructure has long been considered a part of tourism and plays a key role in attracting tourists. Seetanah et al. (2011, p. 92) emphasize "the role of service infrastructure in creating product experience and shaping the overall image of a destination for tourists". Thus, tourism infrastructure is the basis of tourism development. Investment in tourism infrastructure is important in increasing tourist arrivals, and contributes to visitor satisfaction and motivation. As a component of regional tourism, tourism infrastructure is of particular importance for long run tourism growth and the general progress of tourist destinations by providing the required services to tourists. The apparent relationship between tourism development and infrastructure has been confirmed in theory and practice by many authors.

The literature provides different views on the number and type of components representing tourism infrastructure, which can be classified in many various ways. Pearce and Wu (2015) divide tourism infrastructure into two types, namely hard and soft, which Hope (2010, p. 91) called "social and economic infrastructure". According to Enimola (2010, p. 121), "the social infrastructure sub-sector covers some social services like the provision of education, information, town and country planning, health services and other social welfare services in the society"; while "the economic infrastructural sector embraces a group of hard-core economic activities which relate to the production of energy and power, transportation services, water and communication services and others" (Ayodele and Falokun 2003, p. 74). From the model of Pearce and Wu (2015), Bagheri et al. (2018, p. 89) have shown that "to systematize the tourism sector within the soft infrastructure, an amalgamation of diverse factors is shaped, including hospitality, interpretation, and person-to-person encounters that tourists experience". According to Bagheri et al. (2018, p. 89), "Thapa (2012) has also added professional human resources to the sub-set of soft infrastructures, emphasizing the human factor as the most important infrastructure element in developing countries".

Approaching the components of tourism infrastructure, Raina (2005) divides it into four categories, namely: "1. Physical; 2. Cultural; 3. Service; 4. Governance". Ouariti and Jebrane (2020, p. 5) indicated that physical infrastructure includes "hotels, motels, restaurants, transportation, communications, water, electricity"; cultural infrastructure includes "culture, heritage, fairs and festivals, local art and music, dress and dance, language and food"; service infrastructure includes "banking facilities, travel agencies, insurance agencies, tourist guides"; governance infrastructure includes "law and order machinery, customs and immigration".

From the perspective of tourism infrastructure types, Ouariti and Jebrane (2020, p. 5) point out that "the Tourism and Transport Forum (2012) affirms that tourism infrastructure is the supply chain of transport infrastructure, social and environmental infrastructure collaborating at a regional level to create an attractive tourism destination". Among the three components of tourism infrastructure proposed by the Tourism and Transport Forum (2012), social infrastructure is financed mainly by the private sector, while the state mainly controls the environmental and transport infrastructure. The state is responsible

for investing directly in the construction and development of this sector. Today, many countries that want to achieve high business results by attracting more international tourists often focus on increasing investment in the construction and development of tourism infrastructure.

### 2.1. The Role of Transport Infrastructure and Communications Infrastructure

Although scholars approach elements of tourism infrastructure from different perspectives, it is undeniable that transport infrastructure is an important representation of tourism infrastructure and directly impacts the tourism infrastructure that attracts visitors. Kaul (1985, p. 496) stated that "transport plays an important role in the successful creation and development of new attractions as well as the healthy growth of existing ones. Provision of suitable transport has transformed dead centers of tourist interest into active and prosperous places attracting multitudes of people". Indeed, the transport system performs the task of connecting areas with each other, as well as with tourist attractions, and becomes a factor in the competitiveness of the destination. International visitors often go to destinations where transportation systems are available and well developed. Prideaux (2000, p. 53) argues that "if the ability of tourists to travel to preferred destinations is inhibited by inefficiencies in the transport system there is some likelihood that they will seek alternative destinations ". Hence, investment in transport infrastructure development has been an issue of concern for governments for many years.

Along with transport infrastructure, communications infrastructure also plays a vital role in attracting tourists. Communications play an essential role in the development and sustainability of tourism. This helps travelers obtain destination information, make informed decisions about where to go, and helps countries and travel agencies promote and recommend their destinations. Pearce and Wu (2015) indicate that transportation, tourism facilities, and communications are the main components of hard infrastructure. Raina (2005) thinks that traffic and communications are elements in the physical components of tourism infrastructure, along with hotels, motels, and restaurants. Many recent empirical studies have demonstrated the role of transport infrastructure and communications in attracting tourists, resulting in transport infrastructure and communications infrastructure proving to be important factors affecting the number of tourists visiting (Khadaroo and Seetanah 2007b); transport infrastructure is a significant determinant of tourism inflows into a destination (Khadaroo and Seetanah 2008), transport capital having contributed positively to the number of tourist arrivals in both the short-run and the long-run (Seetanah and Khadaroo 2009), the construction of transportation infrastructure promoting the tourism industry (Yu 2016); thus, infrastructure and transportation are important components of the tourism supply chain (Ghaderi et al. 2018); developing transport infrastructures such as highways, airports, and railway stations, has a positive impact on overnight stays in all types of accommodation (Ouariti and Jebrane 2020). Furthermore, Tang (2020) argues that improving transport infrastructure is an important component of trade facilitation and "trade facilitation has improved the efficiency of the inbound tourism market, especially the indicator of infrastructure" Tang (2020, p. 51).

### 2.2. The Role of the Hotel and Restaurant Industry

The hotel industry provides hotel services and organizes short-term accommodation rental services at hotels, campsites, motels, student motels, and guest houses, etc., including restaurant services. In general, the hotel industry provides accommodation and food services for tourists. The hotel and restaurant industry is considered a major component of hospitality and an important components of tourist infrastructure. Hospitality, especially in its commercial incarnation as the "hotel", has emerged as the hub, or the most vital segment, of infrastructure facilities for the travel and tourism industry anywhere around the globe. Raina (2005) considers that, along with transportation, hotels, motels, and restaurants are the physical elements of tourism infrastructure. Meanwhile, the Tourism and Transport Forum (2012) points out that hotels are a significant component of tourism's

social infrastructure and Pearce and Wu (2015) consider them part of the hard infrastructure of tourism.

Like transportation infrastructure, the hotel industry's role (including the restaurant industry) in attracting tourists and developing the tourism industry is evidenced by many recent empirical studies. It is also considered an important component in the tourism supply chain (Ghaderi et al. 2018) and many studies have used rooms as a proxy for tourism infrastructure (Khadaroo and Seetanah 2007b, 2008; Seetanah and Khadaroo 2009; Seetanah et al. 2011; Lim et al. 2019).

*2.3. The Role of Recreation Facilities*

It can be seen that recreational facilities provide attractions, sightseeing, places, and entertainment for visitors during their trip, so is an indispensable component in the tourism infrastructure. Mandić et al. (2018, p. 42) emphasized that "Recreational facilities are an integral part of physical infrastructure which is an indispensable pillar of overall economic and tourism development". Mandić et al. (2018, p. 44) also indicate that "the development of tourism infrastructure and recreational facilities is associated with tourism development (UNWTO 2007; Sharpley 2009)". Adapting the tourism infrastructure model of Jafari and Xiao (2016), Mandić et al. (2018, p. 43) point out that "the physical infrastructure of direct relevance to tourism includes recreational facilities that, along with hotels and other forms of accommodation, spas, and restaurants, form the central tourism infrastructure". In addition, Raina (2005, p. 192) states that "culture and art are also considered elements of the culture which is a component of tourism infrastructure". Therefore, it can be seen that recreational facilities together with transport and communication infrastructure and the restaurant and hotel industries play a part in tourism infrastructure. Each part will promote tourism development by creating attractiveness and enhancing the competitiveness of a destination.

*2.4. The Influence of Uncertain Factors*

According to Vanegas Sr and Croes (2000, p. 951), many qualitative factors influence tourism consumption decisions, such as "special events, political instability, social conflicts, air travel problems, travel restrictions, economic recession and other factors". Typically, dummy variables are introduced to explain the impact of special events that may temporarily affect tourism demand. According to Lin et al. (2015, p. 39), "Greene (2008, p. 106) proposed a dummy variable is a variable that takes the value of one for some observations to indicate the presence of an effect or membership in a group and zero for the remaining observations". Song and Li (2008, p. 217), after reviewing articles on tourism demand modeling and concluded that "researchers should develop some forecasting methods that can accommodate unexpected events in predicting the potential impacts of these one-off events through scenario analysis". Therefore, it can be seen that, in addition to the quantitative variables of investment in tourism infrastructure development, it is necessary to use dummy variables to represent uncertain factors to consider their effects on attracting international visitors.

## 3. Methodology and Data

*3.1. Specification Research Model*

From the literature review, this study hypothesizes that investment in tourism infrastructure such as transport and communication infrastructure, hotel and restaurant industry, and recreation facilities, will positively impact on attracting international visitors to Vietnam, while dummy variables indicate the temporary influence of special events. This relationship is shown by Equation (1) below.

$$VA_{i,t} = f(TC_t, HR_t, EF_t, Dum_{i,t}) + U_{i,t} \tag{1}$$

where $VA_{i,t}$ is the visitor arrivals from source country $i$ in year $t$; $TC_t$ is the capital invested in transport and communications infrastructure in year $t$; $HR_t$ is the capital invested in the

hotel and restaurant industry in year $t$; $RF_t$ is the capital invested in recreation facilities in year $t$; $Dum_{i,t}$ are the dummy variables representing qualitative factors from source country $i$ at time $t$; $U_{i,t}$ is the disturbance term that captures all the other factors that may influence the number of visitor arrivals from source country $i$ at time $t$.

The international visitor arrivals can be divided into several categories, i.e., "sightseeing tourists, business tourists and tourists of other types" (Tang 2020, p. 38) and there can be heterogeneity between them. However, because there are not enough specific data for these objects, heterogeneity between them is not considered.

This study uses regression analysis with a log-log model to estimate the impact of tourism infrastructure development investment on attracting international tourists to Vietnam. In fact, the log-log model is often used to estimate the parameters in order to evaluate the impact level of the independent variable on the dependent variable, because then the effect can be obtained directly from the coefficients (Witt and Witt 1995; Song et al. 2009). Furthermore, the natural logarithmic transformation also reduces data instability (Enders 2004; Studenmund 2006).

There are many techniques to estimate the coefficients of the factors affecting the number of visitors in order to fit the data. Initially, the ordinary least squares (OLS) technique was used commonly for both time series or panel data (such as in the study of Vanegas Sr and Croes 2000; Kulendran and Witt 2001; Lim 2004; Croes and Vanegas Sr 2005; Muñoz 2007). However, OLS regression requires the series to be stationary, otherwise it will lead to spurious regression (Granger and Newbold 1974). One of the technique considered to solve the non-stationary series problem is the cointegration test. The cointegration technique describes "the existence of an equilibrium, or stationary, relationship among two or more time-series, each of which is individually non-stationary" (Banerjee et al. 1994, p. 136). Furthermore, "cointegration techniques permit the estimation and testing of the long-run equilibrium relationships" (Lim and McAleer 2001, p. 1618; Dritsakis 2004, p. 118). Two common estimators for the technique are fully modified ordinary least squares (FMOLS) and dynamic ordinary least squares (DOLS). These estimators need to satisfy one fundamental assumption: the variables included in the models are all non-stationary at level, but stationary at first difference and cointegration of order 1. This technique has been applied in several studies which meet the qualifications (e.g., Dogru et al. 2017). However, these conditions are not always met. Moreover, according to Narayan and Narayan (2005, p. 429), "methods of cointegration are not reliable for small sample sizes". To overcome these limitations, Pesaran and Shin (1999) proposed an ARDL modeling approach. This method is superior regardless of whether the variables exhibit I(0), I(1), or a mixture of both. Song et al. (2003, p. 365) state that "one of the advantages of the general ARDL is that a modern econometric technique, known as error correction, can be readily incorporated into the modeling process". Given these advantages, the ARDL estimation technique has been widely used in recent studies (Song et al. 2003; Lee 2011; Otero-Gómez et al. 2015; Lin et al. 2015; Shafiullah et al. 2018; Kumar et al. 2020).

Based on the above analysis, the nonlinear panel ARDL approach is applied in this study. "Nonlinear ARDL model in panel form which is also a nonlinear representation of the dynamic heterogenous panel data model that is suitable for large T panels" (Salisu and Isah 2017, p. 261). The panel ARDL method also helps to estimate the long-run and short-run relationships for the general sample, as well as the short-run cross-sectional coefficients for each subject, even when the variables are non-stationary and/or show no cointegration. The nonlinear panel ARDL model used in this study is presented in the form of Equation (2) below:

The panel ARDL method also helps in estimation.

$$\Delta lnVA_{i,t} = \mu_i + \sum_{j=1}^{q1} \vartheta_{1ij}\Delta lnVA_{i,t-j} + \sum_{j=0}^{q2} \vartheta_{2ij}\Delta lnTC_{t-j} + \sum_{j=0}^{q3} \vartheta_{3ij}lnHR_{t-j}$$
$$+ \sum_{j=0}^{q4} \vartheta_{4ij}lnRF_{t-j} + \varphi_{oi} + \varphi_{1i}lnVA_{i,t-1} + \varphi_{2i}lnTC_{t-1} + \varphi_{3i}lnHR_{t-1} \quad (2)$$
$$+ \varphi_4 lnRF_{t-1} + Dum_{i,t} + \varepsilon_{i,t}$$
$$i = 1, 2, \ldots N; \ t = 1, 2, \ldots T$$

where $\mu_i$ is the group-specific effect; $i$ is the source country; $t$ is the number of periods (year); $-1 < \varphi_1 < 0$ is the error correction term's coefficient; $\varepsilon_{i,t}$ is the error term; is the first difference operator; j is the lag order decided by the Akaike Information Criterion (AIC); ln is the natural logarithm. For each cross-section, the long-term slope (elasticity) of capital investment in transport and communications infrastructure, the hotel and restaurant industry, and recreation facilities is calculated as $-\frac{\varphi_{2i}}{\varphi_{1i}}$, $-\frac{\varphi_{3i}}{\varphi_{1i}}$, $-\frac{\varphi_{4i}}{\varphi_{1i}}$, respectively, and with the expectation of a positive coefficient. Therefore, the short-term estimate of capital investment in transport and communications infrastructure, the hotel and restaurant industry, and recreation facilities are $\vartheta_{2ij}$, $\vartheta_{3ij}$, $\vartheta_{4ij}$, respectively.

### 3.2. Data

The measurement of tourist attraction to Vietnam in this study is based on international tourist arrivals, as used by many previous studies to measure tourism demand (Khadaroo and Seetanah 2007a; Seetanah and Khadaroo 2009; Seetanah et al. 2011; Mandić et al. 2018). The international visitor arrivals were collected from the ten largest source markets and the remaining markets for 25 years (1995–2019) to form panel data with 275 observations (N = 11 and T = 25). Data on international visitors to Vietnam by source countries in the period 1995–2018 were collected from the VNAT. The ten countries with the most significant number of visitors to Vietnam in the period 1995–2019 are China, Korea, Japan, the United States (US), Malaysia, Australia, the United Kingdom (UK), Singapore, France, and Germany, respectively. These ten source countries accounted for 70.08% of total visitor arrivals to Vietnam from 1995–2019 (Figure 1).

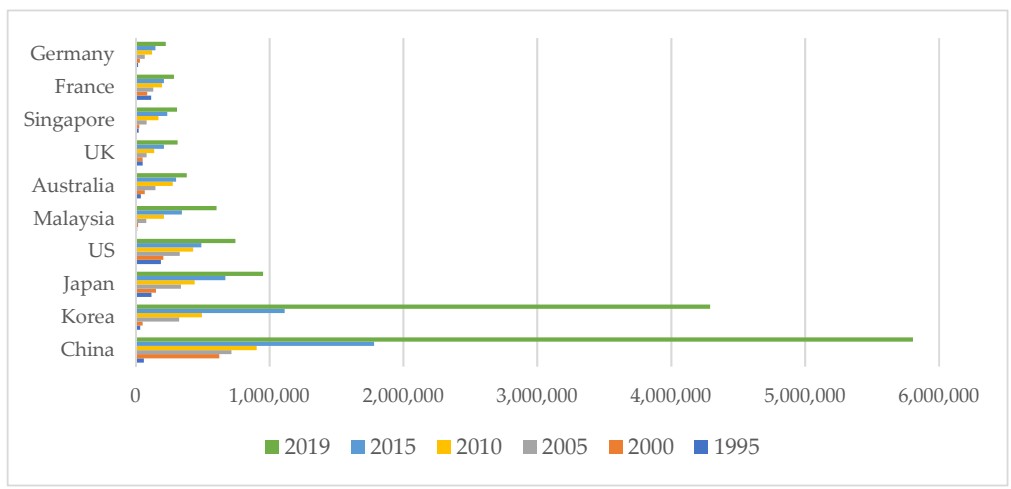

**Figure 1.** Visitors from ten major international markets in the period 1995–2019.

The data series covers 25 years from 1995–2019 and the summary of variables used in the model is described in Table 1 below.

**Table 1.** Summary of variables used in the model.

| Variable | Measure | Description | Data Source |
|----------|---------|-------------|-------------|
| VA | Visitor arrivals | Total number of visitor arrivals per annum | World Tourism Organization (UNWTO) and VNAT |
| TC | Transport and communications infrastructure | Social investment in transport; storage, and communications | GSO of Vietnam |
| HR | Hotel and restaurant industry | Social investment in the hotel and restaurant industry or accommodation, food and beverage service activities | GSO of Vietnam |
| RF | Recreation facilities | Social investment in recreation, culture, and sport or recreation, entertainment, and the arts | GDO of Vietnam |

Note: Data on social investment capital is converted to fixed prices; the original year was 1994.

According to the GSO of Vietnam, the investment capital of the activities in Table 1 for the period 1995–2009 are based on the original year, 1994. However, from 2010–2019 the fixed price is for 2010. Therefore, the fixed price of 2010–2019 is converted to the original year price by the conversion coefficient of the original year 2010 to the original year 1994 according to the Equation (3) below.

$$\text{Conversion coefficient of the original year 2010 to the original year, } 1994 = \frac{\text{Value in year n at the 2010 price}}{\text{Value in 2010 based on the original 1994 price}} \quad (3)$$

Source: Vietnam Ministry of Planning and Investment (2012).

Between 1995 and 2019, there were three years of negative growth in international tourist arrivals to Vietnam: 1998, 2003, and 2009 show −11.4%, 7.6%, and −11.5%, respectively, due to the Asian financial crisis in the late 1990s, the SARS epidemic in 2003, and the global recession in 2008–2009. However, the following year, the number of international tourists to Vietnam increased again and offset previous declines (Figure 2).

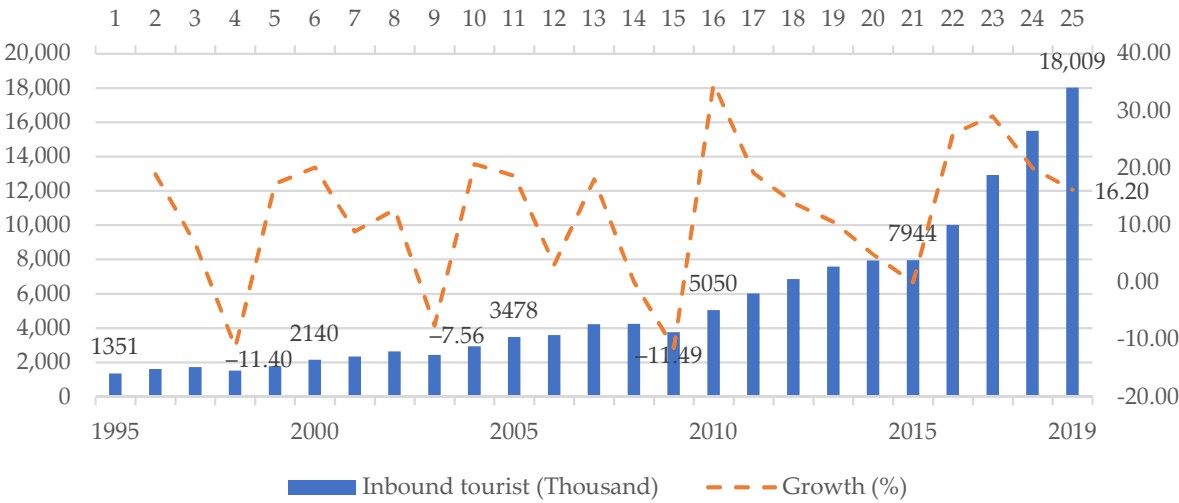

**Figure 2.** Changes in international visitors to Vietnam in the period 1995–2019. Source: Data from UNWTO and VNAT.

Particularly for the Chinese source market, the largest market to Vietnam in recent years, there are also special events such as in 1995, when the relationship between China and Vietnam had not been normalized, so visitors from China to Vietnam faced difficulties obtaining visas; in 2015, China placed an oil rig in Vietnamese waters, straining relations between the two countries and severely affecting tourism. In this study, the above events are considered unstable factors which affected tourists' decision to visit Vietnam. Therefore, the dummy variable used is the value 1, and the remaining cases are assigned the value 0. More details about the methodological use relating to dummy variables can be found in Song and Lin (2010) or Lin et al. (2015). Table 2 below presents descriptive statistics of the variables in the model with 275 observations (11 source markets over 25 years).

**Table 2.** Descriptive statistics variables.

|  | **Ln(VA)** | **Ln(TC)** | **Ln(HR)** | **Ln(RF)** |
|---|---|---|---|---|
| Unit | 1000 person | Billion VND | Billion VND | Billion VND |
| Mean | 12.3131 | 10.3976 | 8.5841 | 8.0096 |
| Maximum | 15.5745 | 11.1848 | 9.5693 | 8.9410 |
| Minimum | 9.5076 | 9.1832 | 7.7227 | 6.7178 |
| Standard Deviation | 1.2666 | 0.6630 | 0.5147 | 0.6452 |
| Coefficient of Variation | 0.1029 | 0.0638 | 0.0600 | 0.0806 |
| Observations | 275 | 275 | 275 | 275 |

Source: Author's calculation using Eviews.

## 4. Research Results and Discussion

### 4.1. The Test Results, Stationarity and Cointegration

Before estimating the parameters, stationarity and cointegration tests were performed to show that the nonlinear panel approach ARDL is appropriate for the data. The unit root test is a popular method for stationary tests for both annual time series and panel data. The stationarity test is conducted in both "individual intercept" and "individual intercept and trend" in test equations. There are many types of unit root test for panel data such as Levin, Lin and Chu t (LLC) and Breitung t-stat with common unit root process; I'm, Pesaran and Shin W-stat (IPS), ADF—Fisher Chi-square (ADF), and PP—Fisher Chi-square (PP) with individual unit root process. The panel data in this study are balanced so that both hypotheses can be applied. The LLC test is chosen for the hypothesis "common unit root process" and the hypothesis "individual unit root process" is chosen for the IPS test. The results of panel unit root tests for logarithms of variables are summarized in Table 3.

**Table 3.** Results of stationarity test.

| | Intercept | | | | Intercept and Trend | | | |
|---|---|---|---|---|---|---|---|---|
| | **LLC** | **IPS** | **ADF** | **PP** | **LLC** | **IPS** | **ADF** | **PP** |
| lnVA | I(1) *** | I(1) *** | I(1) *** | I(1) *** | I(0) ** | I(1) *** | I(1) *** | I(1) *** |
| lnTC | I(0) *** | I(1) *** | I(1) *** | I(1) *** | I(1) *** | I(1) *** | I(1) *** | I(1) *** |
| lnHR | I(1) *** | I(1) *** | I(1) *** | I(1) *** | I(1) *** | I(1) *** | I(1) *** | I(1) *** |
| lnRF | I(0) *** | I(0) *** | I(0) ** | I(1) *** | I(1) *** | I(1) *** | I(1) *** | I(1) *** |

Source: Author's calculation using Eviews. Note: LLC, Levin, Lin & Chu; IPS, I'm, Pesaran and Shin W-stat; ADF, ADF—Fisher Chi-square; PP, PP—Fisher Chi-square; ** and *** for statistically significant at the 0.05 and 0.01 levels, respectively.

According to Table 3, most of the series are non-stationary at level, but stationary at first difference, except for lnVA in LLC test of intercept and trend; lnTC in LLC test of intercept; and lnRF in LLC, IPS and ADF of intercept. Based on the majority of the results, it can be seen that the series are non-stationary at level but stationary at first difference, so a cointegration test should be performed to consider the long-term relationship between variables.

To analyze the cointegration relationship between variables in the panel data model, this study chooses the Pedroni and Kao tests because they are more comprehensive and universal. Cointegration tests are conducted for both "individual intercepts" and "individual intercept and individual trends" in the Pedroni test. By contrast, it is only conducted in the case of individual intercepts in the Kao test. The Pedroni test used seven test statistics (four tests for within-dimension and three tests for between-dimension). The Schwarz Information Criterion (SIC) automatically chooses the lag length with Newey-West automatic bandwidth selection and Bartlett kernel. Table 4 below presents the results of panel cointegration analysis.

**Table 4.** Results of panel cointegration test.

| Method | Statistic | Individual Intercept | Individual Trend and Individual Intercept |
|---|---|---|---|
| Pedroni test | Panel v-Statistic | 1.0575 | 2.8684 |
| | Panel rho-Statistic | −0.7207 | −1.0157 |
| | Panel PP-Statistic | −3.0080 *** | −8.0608 *** |
| | Panel ADF-Statistic | −2.4028 *** | −2.3750 *** |
| | Group rho-Statistic | 0.4850 | 1.1146 |
| | Group PP-Statistic | −3.1699 *** | −5.8950 *** |
| | Group ADF-Statistic | −2.3379 *** | −3.7839 *** |
| Kao test | t-Statistic | −0.7738 | |

Note: *** for statistically significant at the 0.01 levels, respectively; deterministic trend specification: Individual intercept for Pedroni test and Kao test; Four tests for within-dimension of Pedroni test are weighted statistics. Source: Author's calculation using Eviews.

According to the results of the Pedroni test in Table 4, 4/7 tests are significant at the 0.01 level for both "individual intercept" and "individual trend and individual intercept". This means that most cointegration tests in the Pedroni test result in the cointegration series. However, the Kao test gives the opposite result, meaning that the Kao test result does not give cointegration series at the level of 0.05, so is not compelling evidence to conclude clearly that series shows cointegration. Because of lnVA, lnTC, lnHR, and lnRF containing both I(0) and I(1), and when the existence of long-run associations is unclear, the ARDL technique is the most appropriate.

*4.2. Estimated Results*

This study uses the Pooled Mean Group (PMG) estimator to estimate the impact of investment in tourism infrastructure development on attracting international visitors to Vietnam. The PMG estimator is a well-known technique used in the estimation of a dynamic heterogeneous panel data model. Furthermore, by design, in addition to the panel regression results, the PMG also generates results for the individual units (Blackburne and Frank 2007). Thus, computing the impact of tourism infrastructure development on attracting international visitors can assess both long-run and short-run responses for the general sample and each sample (each source market). First, the parameters are estimated by the PMG estimator for the general sample (panel data) with Automatic selection in three maximum lags, Akaike info criterion (AIC) in the Model selection method, and Linear trend in trend specification. Table 5 below summarizes the regression results by the PMG estimator for the general sample for both long-run and short-run.

**Table 5.** Results of regression by the PMG estimator for the general sample.

| Variable | Coefficient | t-Statistic | *p*-Value |
|---|---|---|---|
| Long-Run Equation | | | |
| LnTC | 0.7836 | 4.0925 *** | 0.0001 |
| LnHR | 0.7503 | 7.5976 *** | 0.0000 |
| LnRF | 0.4026 | 3.0775 *** | 0.0028 |
| Dum | −0.3533 | −2.9951 *** | 0.0036 |
| Short-Run Equation | | | |
| COINTEQ01 | −0.4743 | −3.9677 *** | 0.0002 |
| $\Delta \text{LnVA}_{(-1)}$ | 0.1314 | 0.7826 | 0.4361 |
| $\Delta \text{LnVA}_{(-2)}$ | 0.2049 | 1.0379 | 0.3023 |
| $\Delta \text{LnTC}$ | −0.1881 | −0.8512 | 0.3971 |
| $\Delta \text{LnTC}_{(-1)}$ | −0.1081 | −0.5081 | 0.6127 |
| $\Delta \text{LnTC}_{(-2)}$ | −0.3618 | −1.5872 | 0.1162 |
| $\Delta \text{LnHR}$ | −0.2994 | −2.1350 ** | 0.0357 |
| $\Delta \text{LnHR}_{(-1)}$ | −0.3207 | −2.3850 ** | 0.0193 |
| $\Delta \text{LnHR}_{(-2)}$ | −0.3747 | −2.6398 *** | 0.0099 |
| $\Delta \text{LnRF}$ | 0.0073 | 0.0944 | 0.9250 |
| $\Delta \text{LnRF}_{(-1)}$ | 0.3680 | 4.8035 *** | 0.0000 |
| $\Delta \text{LnRF}_{(-2)}$ | 0.2761 | 5.5745 *** | 0.0000 |
| $\Delta \text{Dum}$ | −0.0309 | −0.5882 | 0.5579 |
| $\Delta \text{Dum}_{(-1)}$ | −0.0466 | −1.2999 | 0.1972 |
| $\Delta \text{Dum}_{(-2)}$ | 0.0266 | 0.6770 | 0.5003 |
| C | −2.6244 | −3.8043 *** | 0.0003 |
| @Trend | −0.0059 | −0.6470 | 0.5194 |
| Statistics | | | |

Standard error of regression = 0.0814; Sum squared residual = 0.5565; Log likelihood = 445.9467; Akaike info criterion = −1.8542; Schwarz criterion = 0.6579; Hannan-Quinn criterion: −0.8460. Note: LnVA is dependent variable; ** and *** for statistical significance at 0.05 and 0.01 levels, respectively. Source: Author's calculation using Eviews.

As shown in Table 5, the Log-Likelihood is large; Standard error of regression, Sum squared residual, and Akaike info criterion, Schwarz criterion, and Hannan-Quinn criterion statistics are relatively small, so the model is appropriate and fits with the data. For the long-

run equation, all variables of interest are significant at the 0.01 level, so they are accepted. The estimated coefficients have the same sign as the initial expectation. Investment in tourism infrastructure such as transport and communications infrastructure, the hotel and restaurants industry, and recreation facilities, all positively impact attracting international visitors to Vietnam. Meanwhile, uncertainty factors have been negatively affected.

In the short-term equation, the coefficient of cointegrating equation has a negative sign ($-0.4743$) and is significant at the 0.01 level. This means that the variables converge to the long-run equilibrium, and the convergence rate is 47.43%. The lnTC and Dummy are not significant at the 0.05 level for all lags. By contrast, the variable lnHR is significant at the level, the first difference, the second difference, and lnRF at first difference and second difference, to be more specific, the sign of the coefficients of the negative lnHR and the sign of the positive lnRF coefficients. These findings imply that no significant impact of investment in transport and communications infrastructure has been found on attracting international visitors to Vietnam in the short-term. In comparison, there is a positive effect of investment in recreation facilities, while investment in the hotel and restaurant industry has the opposite effect in the short-run.

Table A1 in Appendix A.1 provides short-run coefficients across cross-sections of the 10 source countries. Accordingly, there are nine source markets moving towards long-run equilibrium, except the US (where the Cointegrating Equation is positive). Additionally, there is at least one coefficient at one level in the short-run of significance at 0.05 or 0.01 for the variables of interest in each source country, except lnTC in the Korean source market. These coefficients indicate the different short-run roles of investments in tourism infrastructure in attracting international visitors to different source markets. At lag 3, the coefficients of lnTC, lnHR, and lnRF are significant in most source markets. Considering this lag, investment in transport and communications infrastructure has different positive and negative roles for each source market in the short-run. To be more specific, investment in transport and communications infrastructure has an active role in source markets in descending order, Germany, the US, Japan, and China. The source markets with a negative role in ascending order are Australia, the UK, France, Malaysia, and Singapore. As for the role of investment in transport and communications infrastructure, investment in the hotel and restaurant industry also has different positive and negative roles for each source market in the short-run. The source markets where it has an active role in descending order are the US, Germany, Japan, respectively. The source markets where it has a negative role in ascending order are Australia, the UK, France, Malaysia, China, and Singapore, respectively. Meanwhile, investment in recreation facilities plays an active role in all source markets. In descending order, these are China, France, Germany, Japan, Korea, the UK, Australia, and the US, respectively. The coefficients of dummy variables with different signs in source markets indicate the short-run impact of different uncertainties on source markets. Positive effects were found in the short-run in China, Korea, Malaysia, Australia, the UK, Singapore, and France. In contrast, the negative effects were found only in Japan, the US, and Germany.

### 4.3. Diagnostic Test and Robustness Check

To further consider the reliability and validity of the model estimate, diagnostic tests are considered. There are two critical diagnostic tests for the panel PMG/ARDL method in Eview: coefficient diagnosis and residual diagnostic. However, according to Wooldridge (2015), based on the asymptotic theory, when there is a sufficient number of observations, it is not necessary to test the normal distribution of the residuals. With 275 observations, this study omits the residual diagnostic and only performs the coefficient diagnostic by coefficient confidence intervals and the Wald test, with the Null Hypothesis that the coefficients are all equal to 0. The results of the diagnostic coefficients are presented in Table 6 below.

**Table 6.** Coefficient diagnostics.

| | | Coefficient Confidence Intervals | | | |
| --- | --- | --- | --- | --- | --- |
| | | 95% Confidence Intervals | | 99% Confidence Intervals | |
| Variable | Coefficient | Low | High | Low | High |
| LnTC | 0.7836 | 0.4029 | 1.1644 | 0.2790 | 1.2883 |
| LnHR | 0.7503 | 0.5539 | 0.9467 | 0.4900 | 1.0105 |
| LnRF | 0.4026 | 0.1424 | 0.6627 | 0.0578 | 0.7473 |
| Dum | −0.3533 | −0.5878 | −0.1187 | −0.6641 | −0.0424 |

Wald test
Null Hypothesis: C(1) = C(2) = C(3) = C(4) = 0
F-statistic: 43.9951 ***; Chi-square = 175.9803 ***

Note: *** for statistical significance at the 0.01 levels, respectively. Source: Results of Wald test.

Table 6 provides the values of the coefficients at the 95% and 99% confidence intervals. Accordingly, the maximum and minimum values of lnTC, lnHR and ln RF are all greater than 0. In contrast, the values of Dummy are all less than 0. The Wald test gives significance at 0.01 level for both F and Chi-squared statistics. Therefore, the null hypothesis is rejected and the alternative hypothesis is accepted, meaning that the estimated coefficients in the model are all non-zero, and they are all necessary for the model. This evidence lends support to the reliability and validity of the estimated model.

Next, the robustness check is performed by comparing the estimated results among PMG/ARDL, cointegration regression and OLS for panel data (assuming the cointegration series from the Pedroni test result). In the OLS method, Random Effects Model (REM) is selected from the Pooled OLS model, Fixed Effect Models (FEM) and REM. In the cointegration regression, the FMOLS estimator is chosen because there is a quite large difference in the long-term coefficient of variance in lnVA (Table 2). The estimated results by FMOLS and OLS methods are detailed in Table A2 in Appendix A.2. The coefficients estimated by PMG/ARDL, FMOLS and OLS methods are compared in Table 7.

**Table 7.** Differences in coefficients estimated by PMG/ARDL, FMOLS and OLS.

| Variable | PMG/ARDL | FMOLS | REM | Difference of PMG with | |
| --- | --- | --- | --- | --- | --- |
| | | | | FMOLS | REM |
| LnTC | 0.7836 *** | 0.7066 *** | 0.7393 *** | 0.0770 | 0.0443 |
| LnHR | 0.7503 *** | 0.5691 *** | 0.5442 *** | 0.1812 | 0.2061 |
| LnRF | 0.4026 *** | 0.0981 | 0.0691 | | |
| Dum | −0.3533 *** | −0.2122 ** | −0.2237 *** | −0.1411 | −0.1296 |

Note: ** and *** for statistical significance at the 0.05 and 0.01 levels, respectively. Source: Estimation results from PMG/ARDL, FMOLS and OLS.

According to Table 7, although the methods produce different estimation results, the signs of the coefficients are similar. To be more detailed, lnTC has quite similar results (bias of no more than 10%), lnHR has a maximum bias of 27.4% and Dummy variable has a bias of no more than 40%. Particularly, lnRF estimated by FMOLS and REM do not reach significance at the 0.05 level. Despite certain differences, it is believed that the results from the PMG/ARDL are more appropriate because of the advantage of PMG/ARDL discussed above, and the cointegration series is still in doubt.

### 4.4. Discussion

The above findings indicate that investment in tourism infrastructure components positively impacts attracting international tourists to Vietnam. In the long-run, increasing 1% of investment capital in transport and communications infrastructure, the hotel and restaurant industry, and recreation facilities will increase international visitors to Vietnam by 0.7836%, 0.7503%, and 0.4026%, respectively. This indicates that capital investment

in transport and communications infrastructure and the hotel and restaurant industry plays a crucial role in attracting international visitors. This evidence lends support to the view that investments in transportation and hotels have played an important role in attracting international tourism, as many earlier studies have found (Khadaroo and Seetanah 2007a, 2007b, 2008; Prideaux 2000; Seetanah et al. 2011). In this study, the role of investment in transport and communications infrastructure (coefficient 0.7836) and investment in the hotel and restaurant industry (coefficient 0.7503) in Vietnam is higher in some areas such as in Mauritius, where the coefficient is found to be 0.36 for investment in transport infrastructure and 0.56 for the investment and hotel industry (Khadaroo and Seetanah 2007b) or 0.32 for investment capital in transport infrastructure and 0.54 for investment and the hotel industry (Seetanah et al. 2011); in 26 island economies, the results are 0.064, 0.16, 0.074 and 0.28 for investment in road, air, communications, and the hotel and restaurant industry, respectively (Khadaroo and Seetanah 2007a); and in 28 countries representing Europe, Asia, America, and Africa, these are 0.13, 0.18, 0.06 and 0.22, respectively, for investment in road, air, port and hotel (Khadaroo and Seetanah 2008). The impact coefficient of the hotel and restaurant industry in this study is lower than that of the hotel accommodation infrastructure in Singapore, from 0.839 to 0.855 in the study by Lim et al. (2019). However, it must also be seen that the different roles of the hotel and restaurant industry depend not only on each country, but also on how the variable that represents it is measured. This role is appropriate because Vietnam is a developing country with great tourism potential and scenic beauty. However, the terrain is difficult, and transportation infrastructure and hotel availability are still limited. With the efforts of the government and the community, the transport and communications infrastructure, as well as the hotel and restaurant facilities in Vietnam, have been significantly improved, creating a favorable environment for tourists, and strongly enticing international visitors to Vietnam. The research results also show that the government and private sector investors cannot expect to see a fast Return on Investment. Their investment in transport and communications infrastructure and hotel and restaurant facilities will only be evident in the long-run. This can be explained by the long lead-in time required by infrastructure works and hotel developments. The impact, therefore, takes time to be fully demonstrated. However, it should be noted that transport and communications infrastructure investment attract visitors and develops other areas of the economy and society, including the hotel and restaurant industry and recreation facilities.

Cross-section short-run coefficients show that, in the short term, the role of investment in the hotel and restaurant industry is decreasing, in this order source markets: the US, Germany, Japan, Australia, the UK, France, Malaysia, China, and Singapore. Meanwhile, the order for investment in transport and communications infrastructure is as follows: Germany, the US, Japan, China, Australia, the UK, France, Malaysia, and Singapore, respectively. This is consistent with the idea that inhabitants of developed countries are accustomed to modern, high-quality transport infrastructure and high-quality restaurants and hotels. Consequently, they prefer to find similar infrastructure in other countries. In contrast, tourists from less developed countries tend to be less demanding of these infrastructures.

Research results also show that investment in recreation facilities is also important to attract international arrivals to Vietnam. Although its role in the long-term is not equal to that of the other two areas of tourism infrastructure in this study, it is effective in both the long-run and short-run. Investment in recreation facilities will directly make destinations more attractive. Formica (2002) states that without attractions, tourism destinations could not exist; attractions are the basis for visitation. These findings are consistent with Vengesayi et al. (2009), suggesting that attractions are the main reason people visit specific destinations and not others. The role of investment in recreation facilities in attracting international visitors in this study is empirical evidence supporting the tourism infrastructure model of Mandić et al. (2018). Accordingly, recreational facilities with hotels and other forms of accommodation, spas, and restaurants form the main tourism infrastructure.

Usually, investment in modern amusement parks will require considerable investment capital. In contrast, investment in developing conservation and ecological tourist areas may require a smaller amount of capital if considered per unit area. The above order of roles of investment in recreation facilities in the short-run implies that in general, visitors want to improve recreation facilities in Vietnam, but visitors from China, France, Germany, and Japan require much more improvement than visitors from Korea, the UK, Australia, and the US. This finding is indicative of visitor preferences from source markets.

## 5. Conclusions and Implications

Attracting international tourists is an essential task for countries as international tourists bring significant income, foreign currency, and jobs to countries, especially potential tourism countries. Therefore, to attract tourists and implement an appropriate pricing policy, investing in tourism infrastructure development to make the destination more competitive and attractive are critical measures. This is the reason why this study examines the impact of investment in tourism infrastructure development on attracting international tourists from empirical research in Vietnam through panel data from 1995 to 2019.

The three types of tourism infrastructure used in this study are transport and communications infrastructure, restaurants and hotels, and entertainment infrastructure. After testing the stationarity and cointegration of the data, this study used the ARDL approach to examine the impact of three tourism infrastructure components on attracting international visitors to Vietnam in the long-run and short-run. In the long-run, investment in tourism infrastructure components has a positive and robust impact on attracting international visitor arrivals. The most decisive impact is investment in transport and communications infrastructure, followed by investment in the hotel and restaurant industry, and finally investment in recreation facilities. However, the short-run impacts of these three types of tourism infrastructure also differ in both sign and magnitude. In addition, different impacts of the three tourism infrastructure components in the short-run on attracting international visitors in general and in each of the leading international visitor markets to Vietnam are also found.

The contribution of this study impinges on two aspects. Firstly, from a theoretical perspective, this study enriches the role of investment in tourism infrastructure in tourism development with three components: transport and communications infrastructure, hotel and restaurant industry, and recreation facilities. Second, from a practical perspective, the study points out the different impacts of components of tourism infrastructure and their specific impact on attracting international visitors to Vietnam as the basis for policies for tourism development.

Overall, investment in transport and communications infrastructure drives economic growth and social development. However, the significant impact on tourism growth in Vietnam revealed in this study justifies the need for government investment in transport infrastructure and information and communications. Besides, investment in the hotel and restaurant industry will provide accommodation, food and beverage services for tourists, especially international tourists. Furthermore, investment in recreation facilities will make the destination more attractive to visitors. Therefore, the positive and vital role of investment in the three tourism infrastructure components is shown in this study because they are the most critical components in the tourism product chain experienced by tourists. On the other hand, although Vietnam has substantial tourism potential, its tourism infrastructure is still limited, so investing in components of tourism infrastructure becomes increasingly pressing to attract visitors in general and international visitors in particular.

Unlike an investment in transport and communications infrastructure that is primarily financed by government funds, investment in the hotel and restaurant industry, as well as recreation facilities, can mobilize the resources of the entire society, especially the private sector, because this is a highly commercialized and profitable sector, and is not prohibitively subject to government control. Thus, internationally and in Vietnam in particular, there is an urgent need for investment incentives for the private sector to help develop these areas.

Although some valuable results have been obtained, this study still has some limitations. Due to data limitations, this study only explores the role of investment in three groups of components without separating each component in detail as well as from different capital sources to see the different roles of the economy sectors. In addition, heterogeneity among visitor arrival groups has not been considered. These issues may provide opportunities for further study.

**Funding:** This research was funded by University of Economics and Law, Vietnam National University, Ho Chi Minh, Vietnam, under grant number 3-2021.

**Institutional Review Board Statement:** Not applicable.

**Informed Consent Statement:** Not applicable.

**Data Availability Statement:** Data on international tourists to Vietnam is collected from various sources, accessible from: https://www.gso.gov.vn/px-web-2/?pxid=V0825&theme=Th%C6%B0%C6%A1ng%20m%E1%BA%A1i%2C%20gi%C3%A1%20c%E1%BA%A3; http://thongke.tourism.vn/index.php/statistic/sub/6; https://www.e-unwto.org/action/doSearch?ConceptID=2473&target=topic. Investment data is available in the GSO statistical yearbooks, accessible from: https://www.gso.gov.vn/du-lieu-va-so-lieu-thong-ke/2020/02/nien-giam-thong-ke-1997/; https://www.gso.gov.vn/du-lieu-va-so-lieu-thong-ke/2020/02/nien-giam-thong-ke-2000/; https://www.gso.gov.vn/du-lieu-va-so-lieu-thong-ke/2020/02/nien-giam-thong-ke-2005/; https://www.gso.gov.vn/du-lieu-va-so-lieu-thong-ke/2019/10/nien-giam-thong-ke-2010-2/; https://www.gso.gov.vn/du-lieu-va-so-lieu-thong-ke/2016/06/nien-giam-thong-ke-2015/; https://www.gso.gov.vn/du-lieu-va-so-lieu-thong-ke/2021/07/nien-giam-thong-ke-2021/.

**Acknowledgments:** The author is grateful to the three anonymous reviewers and academic editor whose comments have contributed to improving the quality of this paper.

**Conflicts of Interest:** The author declares no conflict of interest.

## Appendix A

*Appendix A.1. Result of Cross-Section Short-Run Coefficients*

**Table A1.** Cross-section short-run coefficients.

|  | China | Korea | Japan | US | Malaysia |
|---|---|---|---|---|---|
| COINTEQ01 | −0.2188 *** | −0.5408 *** | −0.1571 *** | 0.2442 *** | −0.9303 *** |
| $\Delta \text{LnVA}_{(-1)}$ | 0.4138 *** | 0.6812 *** | 0.4094 *** | 0.1804 ** | 0.3772 *** |
| $\Delta \text{LnVA}_{(-2)}$ | 0.5513 *** | 0.3598 ** | −0.1618 *** | −0.7762 *** | 0.5502 *** |
| $\Delta \text{LnTC}$ | −1.3162 *** | −0.4403 | 0.2158 *** | −0.0069 | −0.1595 |
| $\Delta \text{LnTC}_{(-1)}$ | 1.6776 *** | −0.2047 | −0.7786 *** | 0.0590 | −0.2536 * |
| $\Delta \text{LnTC}_{(-2)}$ | 0.3185 *** | −0.4225 * | 0.4404 *** | 0.4836 *** | −0.8196 *** |
| $\Delta \text{LnHR}$ | −0.8777 *** | −0.1762 * | −0.0300 *** | 0.2010 *** | −0.1345 ** |
| $\Delta \text{LnHR}_{(-1)}$ | 0.2177 *** | −0.3818 *** | −0.4700 *** | 0.0903 ** | −0.2748 *** |
| $\Delta \text{LnHR}_{(-2)}$ | −0.9323 *** | −0.1601 * | 0.0671 *** | 0.4512 *** | −0.6735 *** |
| $\Delta \text{LnRF}$ | −0.1209 *** | 0.3317 *** | 0.2365 *** | 0.1692 *** | −0.5142 *** |
| $\Delta \text{LnRF}_{(-1)}$ | 0.5206 *** | 0.7553 *** | 0.1835 *** | 0.2281 *** | −0.1185 * |
| $\Delta \text{LnRF}_{(-2)}$ | 0.5467 *** | 0.2941 *** | 0.3317 *** | 0.1021 *** | 0.0653 |
| $\Delta$ Dum | −0.3408 *** | −0.0330 ** | −0.2453 *** | −0.2100 *** | 0.1738 *** |
| $\Delta$ Dum$_{(-1)}$ | −0.2570 *** | 0.0271 | −0.0453 *** | −0.1173 *** | 0.0158 |
| $\Delta$ Dum$_{(-2)}$ | 0.2051 *** | 0.0286 ** | −0.1513 *** | −0.1475 *** | 0.0540 *** |
| C | −1.2348 *** | −3.2720 | −0.7252 ** | 0.9469 | −5.9835 |
| @Trend | 0.0188 *** | 0.0309 *** | −0.0044 *** | 0.0224 *** | 0.0201 *** |

**Table A1.** *Cont.*

|  | **Australia** | **UK** | **Singapore** | **France** | **Germany** |
|---|---|---|---|---|---|
| COINTEQ01 | −0.2230 *** | −0.4722 *** | −1.0421 *** | −0.9949 *** | −0.3443 ** |
| $\Delta$LnVA$_{(-1)}$ | −0.7976 *** | −0.0621 ** | 1.1662 *** | −0.1767 *** | −0.3478 |
| $\Delta$LnVA$_{(-2)}$ | 0.3498 *** | −0.0883 *** | 1.7735 *** | −0.1197 *** | 0.1028 |
| $\Delta$LnTC | 0.5289 *** | 0.3891 *** | −1.7959 *** | 0.0749 * | 0.2754 |
| $\Delta$LnTC$_{(-1)}$ | 0.2878 *** | −0.6483 *** | −0.8821 *** | −0.4528 *** | 0.2184 |
| $\Delta$LnTC$_{(-2)}$ | −0.4132 *** | −0.4869 *** | −1.4587 *** | −0.6643 *** | 0.6033 ** |
| $\Delta$LnHR | 0.0986 *** | −0.1741 *** | −1.4111 *** | −0.3999 *** | −0.0915 |
| $\Delta$LnHR$_{(-1)}$ | 0.1510 *** | −0.5506 *** | −1.3510 *** | −0.5582 *** | −0.4007 *** |
| $\Delta$LnHR$_{(-2)}$ | −0.2002 *** | −0.5204 *** | −0.9590 *** | −0.5994 *** | 0.1700 ** |
| $\Delta$LnRF | −0.3545 *** | 0.0890 *** | 0.0308 | −0.0893 ** | 0.1158 ** |
| $\Delta$LnRF$_{(-1)}$ | 0.2466 *** | 0.4231 *** | 0.6569 *** | 0.3287 *** | 0.6064 *** |
| $\Delta$LnRF$_{(-2)}$ | 0.1928 *** | 0.2773 *** | 0.0310 * | 0.4173 *** | 0.3353 *** |
| $\Delta$ Dum | −0.1033 *** | 0.0528 *** | 0.1058 *** | 0.1854 *** | 0.0245 |
| $\Delta$ Dum$_{(-1)}$ | −0.1194 *** | 0.0688 *** | 0.0447 *** | 0.0486 ** | −0.2367 *** |
| $\Delta$ Dum$_{(-2)}$ | 0.0840 *** | 0.0368 *** | 0.2676 *** | 0.0108 ** | −0.0831 *** |
| C | −0.9987 | −2.6653 ** | −6.2748 | −4.8529 | −2.3403 |
| @Trend | −0.0173 *** | −0.0143 *** | −0.0169 *** | −0.0753 *** | −0.0023 *** |

Note: LnVA is dependent variable; *, ** and *** for statistical significance at 0.10, 0.05 and 0.01 levels, respectively. Source: Author's calculation using Eviews.

*Appendix A.2. Estimated Coefficients*

**Table A2.** Estimated coefficients by FMOLS and OLS.

|  |  | **FMOLS** | **Pooled OLS** | **FEM** | **REM** |
|---|---|---|---|---|---|
| LnTC | Coefficient | 0.7066 | 0.7439 | 0.7392 | 0.7393 |
|  | t-Statistic | 3.067 *** | 1.77 * | 4.78 *** | 4.78 *** |
| LnHR | Coefficient | 0.5691 | 0.5778 | 0.5440 | 0.5442 |
|  | t-Statistic | 5.10 *** | 2.88 ** | 7.38 *** | 7.39 *** |
| LnRF | Coefficient | 0.0981 | 0.0519 | 0.0692 | 0.0691 |
|  | t-Statistic | 0.44 | 0.13 | 0.47 | 0.4708 |
| Dum | Coefficient | −0.2122 | −0.0780 | −0.2247 | −0.2237 |
|  | t-Statistic | −2.06 ** | −0.42 | −3.29 *** | −3.28 *** |
| R-squared |  | 0.9273 | 0.3874 | 0.9204 | 0.8243 |
| Adj R-squared |  | 0.9233 | 0.3783 | 0.9161 | 0.8216 |
| F-statistic |  |  | 42.69 *** | 214.75 *** | 316.57 *** |
| Effects Tests |  |  | 174.10 *** |  |  |
| Hausman test |  |  |  | 0.00 |  |

Note: *, ** and *** for statistically significant at the 0.10, 0.05 and 0.01 levels, respectively. Source: FMOLS and OLS estimation results for data panel in Eview.

Table A2 above shows that adjusted R squared for all estimators is quite high, except Pooled OLS. Significance in F-statistics of Pooled OLS, FEM and FEM are all at 0.01 level. The Redundant Fixed Effects test in the FEM estimate gives significance in the Cross-section F statistic at the 0.01 level, so it allows a strong rejection the null hypothesis that the effects are redundant and shows that the cross-section fixed effects are statistically significant. This means that FEM is more reliable than Pooled OLS estimation. The Chi-square statistical significance of Cross-section random in Hausman Test does not reach 0.05 level, and the Null hypothesis cannot be rejected, so the REM model is selected. These results show that both the FMOLS and REM estimators are reliable.

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
