# Peer review of "Impact of Investment in Tourism Infrastructure Development on Attracting International Visitors: A Nonlinear Panel ARDL Approach Using Vietnam’s Data"

_economies, doi:10.3390/economies9030131_

Round 1
Reviewer 1 Report
Dear Authors,
The manuscript presents interesting research. Congratulations.
In my opinion, it would be worth adding a few literature references from the last 2-3 years.
26.08.2021
Author Response
The author thank the reviewer for the comments that are very helpful to the author to improve the overall quality of the manuscript. The following are the manuscript revisions that I have edited according to the comments of the reviewer 1.
Reviewer’s comment:
The manuscript presents interesting research. Congratulations.
In my opinion, it would be worth adding a few literature references from the last 2-3 years.
Response:
The author has searched and added 4 more documents to the list of references from the last 2-3 years:
Lim,C., Zhu,L., & Koo,T. T. R. (2019). Urban redevelopment and tourism growth: Relationship between tourism infrastructure and international visitor flows. International Journal of Tourism Research,21(2),187-196. doi:10.1002/jtr.2253
Mamirkulova, G., Mi, J., Abbas, J., Mahmood, S. Mubeen, R. & Ziapour, A (2020). New Silk Road Infrastructure Opportunities in Developing Tourism Environment for Residents Better Quality of Life. Global Ecology and Conservation, 20, e01194. doi:10.1016/j.gecco.2020.e01194
Petrova, M., Dekhtyar, N., Klok, O., & Loseva, O. (2018). Regional tourism infrastructure development in the state strategies. Problems and Perspectives in Management, 16(4), 259-274. Retrieved from https://pdfs.semanticscholar.org/0e5f/29d131012ddd160b9e9a1e6464689a788625.pdf
Tang, R. (2020). Does trade facilitation promote the efficiency of inbound tourism? - The empirical test based on Japan. International Journal of Tourism Research, 23, 39-55. doi:10.1002/jtr.2390
promote the efficiency of inbound tourism?—The empirical
TANG 55
Tang R. Does trade facilitation
Besides, contents such as introduction, literature review and discussion have been updated for these references. Specifically, the following paragraphs have been added:
Line 42-44: …Furthermore, the recent studies have shown that tourism infrastructure has a positive impact both directly and indirectly on the quality of life of residents through sustainable tourism development (Mamirkulova et al., 2020).
Line 52-53: …relationship between tourism infrastructure and international visitor flows (Lim et al., 2019)
Line 90-92: …“The maintenance of local tourism infrastructure is becoming an increasingly important prerequisite for the country’s competitiveness” (Petrova et al., 2018, p.259)
Line 101-102: …Further, Lim et al. (2019, p. 187) pointed out that “tourism infrastructure increases tourism demand trends”.
Line 186-189: … Furthermore, Tang (2020) argues that improving transport infrastructure is an important component of trade facilitation and “trade facilitation has improved the efficiency of the inbound tourism market, especially the indicator of infrastructure” Tang (2020, p. 51).
Line 208: … Lim et al., 2019
Line 528-532: … The impact coefficient of the hotel and restaurant industry in this study is lower than that of the hotel accommodation infrastructure in Singapore, found from 0.839 to 0.855 in the study by Lim et al (2019). However, it must also be seen that the different roles of the hotel and restaurant industry depend not only on each country, but also on how the variable that represents it is measured.
Thank you very much for the reviewer's interest and helpful comments
Reviewer 2 Report
This paper investigates the impact of investment in tourism infrastructure components on international visitor attraction using Vietnam data for the period 1995-2019.The paper is very interesting. Structure is clear. Methodology is correct.
Author Response
Response:
The author would like to thank for reviewer's appreciation.
Reviewer 3 Report
With the use of 25 years of data, the analysis in Vietnam just before Covid shows that investing in tourism infrastructure: transport and communications infrastructure, the hotel and restaurant industry, and recreation facilities, has a strong and positive impact on international visitor attraction.
The general approach in identifying the tourism’ different influencing elements is well-structured and thought-through. Even if a lot of other factors could be considered, given the focus of the paper, the processed literature is supporting the major factors. However, some thoughts about the different major tourism types should be mentioned and investigated their relations to the three components of tourism infrastructure. Or at least the paper should reason why tourism is considered as a homogeneous concept and not dealing with its heterogeneity even if the general framework gives implications about.
The justification behind the applied methodology is clear; the statistical tests and their order are followable and according to the general principle of similar investigations.
The results section is excellent with proper interpretation of the results, also giving the same quality of conclusions and implications.
Some minor elements were identified:
In line 45 there is a superfluous semicolon.
In lines 266-267 the quotation should be completed with page indication, and also the ’associates’ canceled from the citation.
In line 271 the sentence seems to be unfinished.
In line 325 the 2011-2019 should be 2001-2019 or in line 324 2001-2019 should be 2011-2019.
Author Response
The author thank the reviewer for the comments that are very helpful to the author to improve the overall quality of the manuscript. The following are the manuscript revisions that I have edited according to the comments of the reviewer 3.
Reviewer’s comment
In addition to the appreciation, the reviewer noted and suggested:
- However, some thoughts about the different major tourism types should be mentioned and investigated their relations to the three components of tourism infrastructure. Or at least the paper should reason why tourism is considered as a homogeneous concept and not dealing with its heterogeneity even if the general framework gives implications about.
- Some minor elements were identified:
In line 45 there is a superfluous semicolon.
In lines 266-267 the quotation should be completed with page indication, and also the ’associates’ canceled from the citation.
In line 271 the sentence seems to be unfinished.
In line 325 the 2011-2019 should be 2001-2019 or in line 324 2001-2019 should be 2011-2019.
Response:
- The author added the reason not to consider heterogeneity between tourist groups.
Please see line 255-258: The international visitor arrivals can be divided into several categories, i.e. "sightseeing tourists, business tourists and tourists of other types" (Tang, 2020, p. 38) and there can be heterogeneity between them. However, because there are not enough specific data for these objects, the heterogeneity between them is not considered.
Also the manuscript has added this issue to the limitations. Please see line 625-626: In addition, heterogeneity among visitor arrival groups has not been considered.
- Some minor elements have been edited according to reviewers
In line 45 (in new version: 47): Remove the superfluous semicolon
In lines 266-267 (in new version: 275): The author edited citation (Banerjee et al., associates, 1994) to (Banerjee et al., 1994, p. 136)
In line 271 (in new version: 281-282): The author has edited it into a complete sentence as follows: This technique has been applied in several studies where they meet the qualifications (e.g., Dogru et al., 2017)
In line 324-325 (in new version: 335-337): The author has edited the period for accuracy.
Thank you very much for the reviewer’s interest and helpful comments
Round 2
Reviewer 1 Report
Well done. Congratulations.
Reviewer 3 Report
The comments and corrections are satisfying the previously identified issues.